# The Analysis of Embryoid Body Formation and Its Role in Retinal Organoid Development

**DOI:** 10.3390/ijms25031444

**Published:** 2024-01-24

**Authors:** Andrea Heredero Berzal, Ellie L. Wagstaff, Anneloor L. M. A. ten Asbroek, Jacoline B. ten Brink, Arthur A. Bergen, Camiel J. F. Boon

**Affiliations:** 1Department of Ophthalmology, Amsterdam University Medical Center (UMC), University of Amsterdam (UvA), Meibergdreef 9, 1105 AZ Amsterdam, The Netherlands; aabergen@amsterdamumc.nl; 2Department of Human Genetics, Amsterdam University Medical Center (UMC), University of Amsterdam (UvA), Meibergdreef 9, 1105 AZ Amsterdam, The Netherlands; e.l.wagstaff@amsterdamumc.nl (E.L.W.); a.l.tenasbroek@amsterdamumc.nl (A.L.M.A.t.A.); j.b.tenbrink@amsterdamumc.nl (J.B.t.B.); 3Emma Center for Personalized Medicine, Amsterdam University Medical Center (UMC), University of Amsterdam (UvA), Meibergdreef 9, 1105 AZ Amsterdam, The Netherlands; 4Department of Ophthalmology, Leiden University Medical Center (LUMC), Leiden University, Albinusdreef 2, 2333 ZA Leiden, The Netherlands

**Keywords:** protocols, embryoid bodies, retinal organoids, retinogenesis, stem cells

## Abstract

Within the last decade, a wide variety of protocols have emerged for the generation of retinal organoids. A subset of studies have compared protocols based on stem cell source, the physical features of the microenvironment, and both internal and external signals, all features that influence embryoid body and retinal organoid formation. Most of these comparisons have focused on the effect of signaling pathways on retinal organoid development. In this study, our aim is to understand whether starting cell conditions, specifically those involved in embryoid body formation, affect the development of retinal organoids in terms of differentiation capacity and reproducibility. To investigate this, we used the popular 3D floating culture method to generate retinal organoids from stem cells. This method starts with either small clumps of stem cells generated from larger clones (clumps protocol, CP) or with an aggregation of single cells (single cells protocol, SCP). Using histological analysis and gene-expression comparison, we found a retention of the pluripotency capacity on embryoid bodies generated through the SCP compared to the CP. Nonetheless, these early developmental differences seem not to impact the final retinal organoid formation, suggesting a potential compensatory mechanism during the neurosphere stage. This study not only facilitates an in-depth exploration of embryoid body development but also provides valuable insights for the selection of the most suitable protocol in order to study retinal development and to model inherited retinal disorders in vitro.

## 1. Introduction

Advanced cell culture methods have become an essential tool in vision research. While two-dimensional culture models provide invaluable information, they do not resemble the complex in vivo architecture of tissues [1]. Since the emergence of stem cell-derived 3D retinal organoids (ROs) in 2009 [2], these three-dimensional systems have replaced the more traditional two-dimensional in vitro models. The application of human embryonic stem cell (hESC) and human induced pluripotent stem cell (hiPSC) technology to generate ROs has overcome a number of limitations. ROs represent many key aspects of retinal development, including the generation of most of the retinal-specific cell types, their proper localization resulting in characteristic lamination, and the capacity to recapitulate human retinogenesis [3,4,5]. Nonetheless, current ROs still lack the cellular diversity and complexity found in vivo [6,7].

A great effort has been made to improve and adapt protocols to generate ROs to address unique research questions, resulting in a wide variety of protocols. However, these different methods of generating ROs still present multiple challenges, such as stem cell line-dependent heterogeneity, due to cell type conversion and epigenetic memory of hESCs and hiPSCs [8,9]. Other sources of variability include subtle differences between protocols, different technical experiences of personnel, and/or technical preferences of laboratories [10]. These differences are difficult to address systematically given the labor-intensive and time-consuming nature of the protocols [9]. This highlights the need for a deeper insight into the stages and events that modulate RO formation.

Three categories of parameters controlling RO development have been defined previously: (1) the physical features of the culture environment; (2) the need for endogenous and exogenous signals; and (3) the starting material and conditions for embryoid body (EB) formation [11]. Investigative reports that have compared these categories agree that a systematic comparison, examining specific parameters within the same cell line, is necessary to establish definitive conclusions [10].

In order to increase our understanding of RO development, we previously reviewed the wide range of accessible organoid protocols [12]. This analysis revealed that 40% of the RO generation protocols start with stem cell clumps in suspension, followed by protocols that begin using single-cell aggregation methods (approximately 35%) [12]. Both stem cell clumps and single-cell aggregates are then cultured to self-organize into EBs, capable of mimicking various aspects of embryogenesis [13,14,15]. Building on that, we further evaluated articles that systematically review the different parameters within RO development and created an overview based on the three aforementioned key parameters (physical features of the culture environment, exogenous and endogenous signals, and starting material and conditions for EB formation). The starting material and conditions for EB formation plus their influence on the initial development of both EBs and ROs have not been studied as extensively as the physical cues of the culture environment and the exogenous/endogenous signals. However, evidence suggests that the method of EB formation does impact internal cell mechanics, affecting the internal cellular mechanism, including cell–cell contact, cell–matrix interactions, germ layer specification, and the activation of molecular pathways, and thus, potentially RO formation [15].

In this paper, we utilized one single hESC line (commercially available WA01 hESC), and we assess the impact of EB formation methods on RO development. We selected two different protocols: a stem cell clump protocol (clumps protocol, CP) [16], and a single-cell aggregation protocol (single cells protocol, SCP) [17], due to their widespread use for RO generation, and systematically evaluated the follow-up events. This approach allows us to investigate the effects of initial cell mechanics on EB formation and, eventually, its influence on RO development.

## 2. Results

### 2.1. Comparison of the Events That Influence Retinal Organoid Development in Previously Published Systematic Reviews

We evaluated the most recent literature for ROs derived from human tissue. We identified seven papers that systematically examined developmental stages during RO development. We classified these reports into three categories: the physical features of the culture environment, the use of exogenous signals, and the starting cell conditions. A comprehensive summary of this analysis is provided in Table 1. We found three reviews that systematically provided information on the effect of physical features of the culture environment [9,18,19]. The impact of exogenous signals was investigated in three other independent studies [10,12,20]. The starting cell culture conditions were evaluated by Mellough and colleagues (2019) [8]. This single report emphasizes the current under-evaluation of the impact of starting cell conditions on RO development.

### 2.2. Morphological Characterization Reveals Differences between Protocols at the Embryoid Body Stage

The hESC line (WA01) used in this study was dissociated into stem cell clumps using EDTA (following the CP) or into single cells using Accutase (following the SCP). In order to ensure cell survival, the SCP also adds Rho kinase inhibitor Y-27632 (ROCKi) after single-cell dissociation. Therefore, the CP and SCP exhibit variations in initial cell conditions, facilitating the study of the effect of a non-forced (CP) and a forced (SCP) aggregation process on EB formation. At the same time, events such as EB plating, neurosphere scraping, RO separation, and supplement additions (CHIR99021, Fetal Calf Serum (FCS), Taurine, Retinoic Acid (RA), and Triiodo-L-Thyronine (T3)), as well as the timing of such events, are consistent across both protocols (Figure 1A). Therefore, after the post-embryoid body formation stage, the subsequent steps are shared between CP and SCP. However, modifications were made to the original protocols; additional details are provided in the Materials and Methods section. We collected samples from the four primary representative stages of the RO development process: hESCs (day 0), EBs (days 4 and 7), neurospheres, (day 18), and ROs (days 43 and 120), with particular emphasis on the evaluation of the initial EB stage(s) (Figure 1B) to increase our knowledge of the early events that guide the RO differentiation process.

In order to make the CP and SCP more comparable, we first investigated the initial cell conditions of EB formation and size via the CP. These EBs presented heterogeneity in shape, with an average diameter of 237.5 ± 52.36 μm at day 7 (Figure 1C). To achieve similar-sized EBs using the SCP, we adjusted the initial cell seeding density to 250 cells per well at day 0. This resulted in highly homogeneous EBs for both shape and size, with an average diameter of 235.7 ± 42.23 μm on day 7 (Figure 1C). This initial optimization step to adjust the EB size became crucial in this study due to the well-known variability associated with EB size. It was shown that oxygen transport was reduced in large EBs (>300 μm diameter) [21]. Although this factor alone does not alter the differentiation potential, more studies have reported size as a key parameter to control in differentiation, since small EBs are prone to form endoderm, while larger EBs generate mesoderm [22].

EBs develop cells representing all three germ layers: endoderm, mesoderm, and ectoderm in vitro [23]. However, their arrangement appears less organized compared to early in vivo development [24]. At this early developmental stage, we observed differences between the two aforementioned protocols: while no apparent morphological development was evident in the CP, EBs generated via the SCP displayed a primitive endoderm phenotype (Figure 1B). Following the formation of the primitive endoderm, observed within 2 days in vitro, cavities began to develop within the core of these EBs. The process of cavity formation appeared heterogeneous within EBs, as they displayed either a single large cavity or multiple small cavities (Figure 2, bottom). In contrast, EBs produced using the CP did not exhibit any indications of primitive endoderm formation or cavitation under brightfield microscopy (Figure 1B,C).

On day 7, EBs from both protocols were plated and maintained in a 2D environment until day 18. During this phase, both protocols successfully produced healthy neurospheres, characterized by the formation of self-formed ectodermal autonomous multi-zones, as previously described by Isla-Magrane and colleagues (2021) [25]. These zones are able to produce colonies containing various ocular lineages [26]. A detail of the neurospheres (day 18) generated via both protocols can be seen in Appendix A. Neurospheres are also referred to as optic vesicle-like structures or self-organized multi-zone ocular progenitor cells in more recent papers [9,25].

For a comprehensive understanding of EBs, it is crucial to examine their entire structure without sectioning. The process of sectioning might potentially damage the tissue and disrupt the overall structure of EBs or ROs, as mentioned by Berber and colleagues (2022) [27]. We therefore conducted staining on intact EBs for the pan-neuronal marker β-III Tubulin. Notably, EBs generated using the CP exhibited less densely packed β-III Tubulin, displaying a dendritic distribution pattern within the EB. In contrast, EBs generated via the SCP showed exclusively β-III Tubulin expression in a tightly compacted layer at the outer perimeter of the EB. This observation confirmed the presence of cavities within the core of the SCP-derived EBs, as already visualized under brightfield microscopy (Figure 1B,C).

### 2.3. Single Cells Protocol Generates Embryoid Bodies That Delay the Epithelial-to-Mesenchymal Transition

We assessed the transition from pluripotent stem cells toward retinal fate in both protocols by collecting samples at days 0, 4, 7, 18, and 43. We also conducted more detailed analyses at the EB stage to assess whether the observed morphological differences could be correlated with alterations in gene-expression levels.

EBs generated using the SCP exhibited a primitive endoderm-like phenotype. Subsequently, we examined this potential structure using markers to evaluate not only the formation of embryonic tissues but also their organization during EB development. For this purpose, we used GATA4 (a primitive endoderm marker) and SOX2 (a primitive ectoderm marker). By day 4, EBs generated using both protocols showed upregulation of GATA4, maintaining its expression within the first developmental week (Appendix A). In contrast, SOX2 remained absent on days 4 and 7 for both protocols (Appendix A). These results suggest the presence of a primitive endoderm phenotype-like expression in EBs generated using both protocols.

Further analysis was conducted to detect developmental differences within the EB stage and later on in development. We therefore assessed the temporal gene expression of the pluripotency gene *NANOG* and the neural progenitor *VSX2* via qPCR (Figure 3A,B). Interestingly, EBs generated using the SCP showed significant upregulation of *NANOG* by days 4 (*p* < 0.0001) and 7 (*p* < 0.0001) compared to the CP (*n* = 3), followed by a decrease in its expression by days 18 and 43, in line with the expected transition toward a differentiated stage. qPCR analysis of the neural progenitor *VSX2* gene revealed robust expression in both groups at day 18, increasing its expression by day 43, where no significant differences were detected between protocols (Figure 3B).

To confirm the potential retention of the pluripotency capacity on EBs generated through the SCP, we next examined whether the EBs of both protocols were composed of undifferentiated hESCs or differentiated cells. The formation of EBs from stem cells is possible because of the adhesion molecule E-cadherin (CDH1), which is strongly expressed in undifferentiated hESCs [28]. Upon differentiation, E-cadherin expression is turned off, and N-cadherin (CDH2) expression is switched on, a key event during the epithelial-to-mesenchymal transition (Figure 4A). We observed differences in expression of CDH1 and CDH2 in EBs that are in line with the observed morphological differences between protocols by days 4 and 7 under bright field microscopy, described above. EBs generated using the CP exhibited a scattered CDH1 distribution by day 4. In regions where CDH1 expression decreased, a simultaneous increase of CDH2 expression was observed (Figure 4B). The E- to N-cadherin switch became more pronounced by day 7 (Figure 4C) when a significant reduction of CDH1 expression, now localized only in small regions within the EB, was accompanied by a robust and uniform distribution of CDH2. In contrast, EBs generated using the SCP showed robust and uniformly distributed CDH1 expression throughout the EB sections, with complete absence of CDH2 expression on day 4 (Figure 4B). This pattern was also present on day 7 (Figure 4C).

These results were also consistent with the gene-expression evaluation of *CDH1* and *CDH2* using qPCR (Figure 4D,E), where we also evaluated later timepoints (days 18 and 43). In EBs generated via the CP, the downregulation of *CDH1* started already by day 7, with complete absence by day 18. In contrast, in EBs generated through the SCP, *CDH1* expression showed a significant upregulation by day 7 (*p* < 0.001). By day 18, the SCP EBs had downregulated *CDH1* expression, although significant differences were detected between protocols (*p* < 0.001), followed by complete absence by day 43 (Figure 4D). *CDH2* expression exhibited the opposite pattern to *CDH1*. EBs generated via the CP presented a significant upregulation by day 4 (*p* < 0.001) and day 7 (*p* < 0.001) compared to the EBs generated via SCP. This highly upregulated expression was sustained thereafter on days 18 and 43. In contrast, EBs generated using the SCP showed low levels on days 4 and 7, with a delayed upregulation observed by day 18 (Figure 4E).

Taken together, our results confirm that EBs produced via both protocols undergo complex morphogenesis processes during differentiation, characterized by the emergence of genes linked to the epithelial-to-mesenchymal transition. However, we also found that in EBs generated through the SCP, this transition appears to be potentially delayed compared to the CP.

Despite the retained pluripotency observed in EBs generated using the SCP, *CDH2* levels gradually increased throughout retinal organoid development, eventually reaching levels comparable to those observed in the CP by day 18 (Figure 4E). A consistent pattern was observed for *NANOG* (Figure 3A) and *VSX2* (Figure 3B) where, despite initial variation at the EB stage, the neurosphere stage seems to include a potential compensatory mechanism, resulting in comparable expression levels for both protocols by day 18.

These findings highlight the evident progression from pluripotency toward the eye field, regardless of the methodology chosen. However, differences were observed between the two protocols in the early stages, suggesting a retention of the pluripotency capacity in EBs generated using the SCP.

### 2.4. Early Differences during Embryoid Body Development Do Not Interfere with Retinal Organoid Development

The transition from pluripotent stem cells to the eye field was evaluated by assessing the expression of the key regulators at the RO stage for early development (day 43) and late development (day 120). For this evaluation, we used immunohistochemistry to detect differences in the architecture and organization of the retinal layers.

Both protocols generated ROs at day 43 with high heterogeneity in shape and size but presenting the characteristic neuroretina. No differences were detected between the protocols upon brightfield microscopy evaluation (Figure 5A).

Next, we evaluated whether ocular developmental regulators were affected by pluripotency retention within the EB stage of the SCP compared to the CP. To address this, we selected the retinal homeobox protein RAX and the early photoreceptor progenitor RCVRN, investigating their corresponding protein localization in ROs at day 43. Immunohistochemistry evaluation did not reveal significant differences. RAX was robustly expressed within the neuroretinal regions, presenting similar architecture in all ROs (Figure 5B). The same organization was also revealed by RCVRN being present in the outer layer of the neuroretina (Figure 5B). After cellular quantification, ROs at day 43 presented similar levels of cells expressing RAX and RCVRN (Figure 5D,E).

Next, we evaluated the neural progenitor VSX2 marker and the photoreceptor-specific transcription factor CRX (cone–rod homeobox) at day 43 (Figure 5C). These proteins were robustly expressed in RO sections generated using both protocols. These immunohistochemistry evaluations also reveal similar structural organization, generating the neuroretina as a compacted layer, where both markers presented a robust expression. Cellular protein staining quantification for CRX in ROs at day 43 did not reveal differences between protocols (Figure 5F).

By day 120, ROs generated from both protocols increased in size, and in both cases ROs presented the protruding outer segments, as a preliminary stage to the formation of the full outer segment structure (Figure 6A). Regarding photoreceptor-specific markers, the rod-specific rhodopsin (RHO) and the cone-specific red–green opsin (R/G opsin) did not reveal differences in the structural organization (Figure 6B) or cell quantification (Figure 6D,E), aligning with the results already discussed for day 43. ROs cultured long-term with both protocols exhibited similar levels of expression and distribution of RCVRN and CRX, as confirmed by the results shown in Figure 6C. This confirmation was further supported through cell quantification, as depicted in Figure 6F,G.

## 3. Discussion

ROs have gained considerable attention in the fields of developmental research, gene therapy, and personalized medicine research approaches. Nonetheless, the currently available protocols for RO generation are highly variable, presenting a wide variety of factors that can impact developmental reproducibility, differentiation efficiency, and RO yield. An observed increase in the number of publications comparing RO protocols emphasizes the interest in understanding the individual events that differ between protocols. An initial analysis of these comparative studies was possible, and they could be separated into three main areas: physical features of the culture environment, exogenous and endogenous signals, and starting material and conditions for EB formation. This initial evaluation identified just one study that compared EB formation methods, and its impact on RO development [8]. However, evidence suggests that EB formation influences internal cellular mechanisms, including cell–cell contact, cell–matrix interactions, germ layer specification, and the activation of molecular pathways, among others [15]. These factors ultimately affect cell fate decisions and, as a consequence, the differentiation capacity. This context emphasizes the need for further research into the EB developmental stage. The selected protocols (CP and SCP) gave us the opportunity to focus on the early stages and the differences that reside in the EB formation method. The CP used EDTA to dissociate stem cells into clumps, while the SCP used Accutase to dissociate stem cells into single cells, which later on were supplemented with ROCKi to ensure cell survival. Although the selection of EDTA and Accutase may introduce variability, the use of these factors helps us to obtain the desired degree of dissociation to generate clumps or single cells, respectively.

Therefore, in this study we investigated the effect of EB formation methods on EB and RO development. Our findings suggest that EBs generated according to the described SCP are able to maintain pluripotency for an extended period compared to EBs generated using the CP. Interestingly, these differences diminish throughout development, due to a potential compensatory mechanism occurring during the 2D stage, leading to developmentally similar neurospheres by day 18. As a consequence, these initial differences detected in the EB stage do not cause any adverse effects within RO development, in terms of gene expression and retinal architecture.

We selected two of the most-used methods for generating ROs, ensuring that both protocols present the same timeline of the main events [12]. This allowed us to systematically focus on the EB formation method and its potential consequences for RO development. Noticeable morphological differences were already observed under brightfield microscopy within 48 h of differentiation. Zeevaert and colleagues (2020) previously reported an epithelial outer layer surrounding the EBs that resembles the embryonic primitive endoderm [15]. During embryonic morphogenesis, the primitive endoderm-like phenotype has been described as a preliminary process before the formation of cystic cavities occurs [29]. Both the organized primitive endoderm-like phenotype and the cavity formation process were characteristics that we observed in EBs generated via the SCP. In contrast, EBs generated via the CP present indications of primitive endoderm expression in a disorganized manner, with no signs of cavitation. In the proposed mechanism of early human development, as described by Pedroza et al. (2023), single human pluripotent stem cells start the self-assembly step, followed by the self-sorting and self-patterning steps [30]. This sequence of events allowed the early specification of epiblast and hypoblast and a subsequent patterning similar to the one observed in EBs generated via the SCP. This first observation highlights the SCP’s ability to generate EBs that follow a developmental path resembling natural human development. Nonetheless, a further investigation of structures such as the mesoderm and basement membrane would be necessary to understand to what extent the SCP-derived EBs can mimic natural human development.

We found that EBs generated using the SCP exhibited a prolonged retention of pluripotency compared to those generated using the CP. The pluripotency gene *NANOG* displayed significant upregulation for EBs generated using the SCP on both days 4 and 7, suggesting the absence of early differentiation processes in these primordial EBs. Sustained differences were observed as early as day 4, becoming more pronounced by day 7. Consistent with these findings, the temporal gene-expression analysis of EBs generated using the CP revealed the initiation of the epithelial-to-mesenchymal transition process by day 4, a prerequisite for progressing toward differentiation. In contrast, EBs generated under the SCP showed non-indications of the epithelial-to-mesenchymal transition, and this process did not begin during the EB stage in the SCP. The epithelial-to-mesenchymal transition provides an indication of the developmental stage of the EBs, since undifferentiated hESCs highly express E-cadherin [28]. These observations are in line with our hypothesis, since the SCP uses a supplementation of ROCKi, known to promote cell survival while maintaining pluripotency [31], during the initial steps of differentiation. Single cells require ROCKi supplementation in order to manage successful EB formation. ROCKi is also known to heavily regulate the actomyosin-driven forces that are crucial for self-organization [32], which could explain the self-organization observed in EBs generated using the SCP. Additionally, ROCKi supplementation leads to the stabilization of E-cadherin [33], which is consistent with the observed results in our study. As a consequence of this stabilization, SCP-derived EBs do not undergo mesenchymal transition, showing a retention of the pluripotency capacity compared to CP-derived EBs. Previous studies have reported the need for further investigation into the impact of ROCKi on gene expression [8,34]. While the primary focus of this study is not on the isolated effects of ROCKi, we have presented a compilation of genes that could potentially be influenced by the supplementation of ROCKi.

Gene-expression evaluation demonstrated that, eventually, *NANOG*, *VSX2*, and *CDH2* reach comparable levels in neurospheres generated under both protocols. This suggests a potential compensatory mechanism in the SCP that is guiding the differentiation process. In previous studies comparing suspension and adherent 2D protocols, it was demonstrated that this step had a direct impact in the developmental timing of ROs, as well as in the organization of retinal layers [9]. In this study, it was suggested that the adherent culture method generated ROs with well-organized layers of the inner and outer retina. The authors attributed this organization to the use of Matrigel as a coating medium for growing neurospheres during the 2D stage. Matrigel generates a gradient of factors, helping in the development of laminated ROs, with the authors claiming that in the absence of these factors, ROs developed in a disorganized fashion. However, in our selected protocols, EBs were plated with a fetal calf serum coating, and both protocols generated well-organized ROs with similar lamination comparable to the one shown by Radojevic and colleagues (2021) [9]. Therefore, the effects observed in that study may not be attributed to the protein content of Matrigel, but rather to the ability to grow in a 2D environment. In our study, long-term ROs did not show variability in organization and lamination when comparing the CP and SCP, confirming a consistent pattern in development. By day 120, we observed the presence of photoreceptors in ROs generated from both protocols, including expression of RHO and R/G opsin, indicating successful maturation using both methods.

Only one study has previously evaluated the effect of EB formation methods upon RO development [8], which found that EB formation methods, as well as maintenance conditions, had a lasting effect throughout RO development. Here, we suggest that the influence of maintenance conditions outweighs the impact of the EB formation method in determining the differentiation capacity of ROs. Despite the initial differences observed in EBs, the subsequent 2D stage allows them to undergo proper development during differentiation. Nevertheless, it is important to note that although similar protocols were evaluated by Mellough and colleagues (2019) [8], the methods used for colony detachment or the seeding densities used were different than the conditions evaluated in our study.

Our findings lead to the conclusion that the SCP-derived EBs retain pluripotency capacity, therefore highlighting that the EB formation method does indeed affect early development. This finding can be crucial when modeling congenital or early-onset inherited retinal dystrophies (IRDs), where key genes undergo upregulation during the early stages of development. It is essential to take these findings into account when selecting the most suitable protocol, as it can significantly impact the final outcome of the research. However, we also conclude that the early differences observed between the methods of EB formation do not interfere significantly with the development and maturation of ROs. Therefore, for IRDs in which key genes present upregulation in late stages, it is not the EB formation method, but rather the maintenance conditions as previously reported by Mellough et al. (2019) that may strongly influence the timing of gene-expression onset of the genes of interest [8].

## 4. Materials and Methods

### 4.1. hESC Culture

We used the commercially available WA01 hESC line (WiCell, Madison, WI, USA). The cells were maintained on hESC-qualified Matrigel (Corning, New York, NY, USA) coated well with mTeSR Plus medium (STEMCELL Technologies, Seattle, WA, USA) and were passed twice a week using 0.5 mM EDTA (Invitrogen, Waltham, MA, USA) in 1× DPBS (Thermo Fisher Scientific, Waltham, MA, USA).

### 4.2. Description of Embryoid Body Formation Protocols

Clumps protocol (CP)—EB production (adapted from Ohlemacher et al., 2015 [16]). Briefly, hESCs were expanded to reach approximately 70% confluency at the beginning of the induction and detached with 0.5 mM EDTA in 1× DPBS. Dissociated stem cell clumps were transferred to a 15 mL falcon tube and centrifuged for 30 s at 140 relative centrifugal force (rcf). Cell clumps were resuspended in a 3:1 mixture of mTeSR1 (STEMCELL Technologies) and NIM (neural induction medium), containing DMEM/F12 (1:1 Thermo Fisher Scientific), 1% N2 supplement (Thermo Fisher Scientific), 1% NEAAs (Sigma, St. Louis, MO, USA), 2 µg/mL heparin (Sigma), 1% penicillin/streptomycin (Thermo Fisher Scientific), and 1% Glutamax (Thermo Fisher Scientific) to induce EB formation, and then were transferred to a T25 flask (Corning). This was noted as day 0. Stem cell clumps were gradually transitioned to NIM according to the following scheme: day 0 (3:1 mTeSR1:NIM), day 1 (1:1 mTeSR1:NIM), day 3 (1:3 mTeSR1:NIM), and day 4 (NIM). After 7 days in floating culture, the EBs were plated onto uncoated 6-well plates and cultured in NIM. For the first 24 h, 10% (*v*/*v*) heat-inactivated FCS (fetal calf serum) (Thermo Fisher Scientific) was added to facilitate EB attachment. In general, approximately 30–40 EBs were plated per well, with NIM refreshed every other day.

Single cells protocol (SCP)—EB production (adapted from Huang et al., 2019, with modifications [17]). Briefly, hESCs were expanded to reach approximately 70% confluency at the beginning of the induction and following one washing step with 1× DPBS were dissociated into single cells with Accutase (Thermo Fisher Scientific) incubation for 5 min at 37 °C. Single cells were centrifuged at 200 rcf for 4 min, before the pellet was resuspended in mTeSR1 with 10 µM Rho kinase inhibitor Y-27632 (ROCKi) (SelleckChem, Houston, TX, USA). Cells were seeded at 250 cells per well of a U-bottom 96-well plate (Corning), followed by a centrifugation step at 200 rcf for 4 min. They were maintained in a 3:1 ratio of mTeSR1 and NIM as described above. Cells were gradually transitioned according to the following scheme: day 0 (3:1 mTeSR1:NIM), day 3 (1:1 mTeSR1:NIM), day 5 (1:3 mTeSR1:NIM), and day 7 (NIM). EBs from the entire 96-well plate were transferred to a single well of a 6-well plate and cultured in NIM with 10% (*v*/*v*) heat-activated FCS.

Retinal organoid differentiation (the following steps are shared between both protocols.) On day 16, the medium was changed from NIM to retinal differentiation medium (RDM, modifications are made to the original protocols) containing DMEM/F12 (1:1), 2% B27 supplement with vitamin A (Life Technologies, Carlsbad, CA, USA), 1% NEAAs, penicillin/streptomycin, and 1% Glutamax. At day 18, neurospheres were fully formed and ready to be scraped off with the checkerboard scraping method, as described by Cowan et al. (2020) [35], which allows breaking the tissue without breaking the neurosphere structures. This has been modified from the original method, since both protocols dislodge the neural cluster by flushing with a P1000 pipette. However it was reported that manual selection was highly dependent on technical skills, therefore introducing more variability [9]. The excised neurosphere clumps were transferred to a 15 mL falcon tube (Greiner, Kremsmünster, Austria) and washed 3 times with RDM, allowing the clumps to sink between washing steps. The neurosphere clumps were transferred to a 60 mm non-treated culture dish (Eppendorf, Hamburg, Germany) for long-term culture. CHIR99021 (3 µM) (Merck Millipore, Burlington, MA, USA) was supplemented between days 18 and 24 (this was an addition to the original protocols). The RDM medium was changed every other day. Between day 25 and 30, ROs (post-differentiation) were sorted from non-retinal organoids. For long-term suspension culture, for both protocols, the medium was supplemented with 10% FCS and 100 µM Taurine (Sigma) beginning on day 35. After a week, Retinoic Acid (1 µM) (RA) (Sigma) was added. On day 84, Triiodo-L-Thyronine (40 ng/mL) (T3) (Sigma) was added, and RA concentration was halved. T3 addition and RA reduction were modifications made to the original protocols to promote well-stratified photoreceptor maturation.

To compare protocols, samples were collected from both methods at four developmentally distinct stages: hESCs (day 0), the EB stage (days 4 and 7), the neurosphere stage (day 18), and the RO stage (day 43 and day 120).

### 4.3. Tissue Preparation and Immunohistochemistry

Immunohistochemistry sections were fixated in 4% PFA (Sigma) in 1X PBS (Thermo Fisher Scientific) for 20 min; washed three times for 5 min in 1X PBS; then incubated for 30 min in 1X PBS containing 15% sucrose (Merk, Rahway, NJ, USA), followed by 30 min in 1X PBS containing 30% sucrose; and embedded in Tissue-Tek O.C.T. (Sakura Finetek, Torrance, CA, USA) on dry ice and cryosectioned in 10 μm sections for ROs and 6 μm sections for EBs onto Superfrost Plus slides (Thermo Fisher Scientific) coated with Poly-l-lysine (Sigma) and stored at −20 °C.

Sections were washed three times for 5 min with 1X PBS. Primary antibodies (Table 2) were applied for 2 hr at 4 °C in 1X PBS containing 10% BSA (Sigma) and 10% Triton X-100 (Merk). Following three washes with 1X PBS for 5 min, secondary antibodies (Table 3) were applied for 1 hr at room temperature in 10% Triton X-100 and 1 µg/mL DAPI (Sigma). After three washing steps with 1X PBS, the sections were mounted using ProLong Gold Antifade Mountant (Thermo Fisher Scientific).

For whole-EB staining, EBs were stained following a previously described method [10]. Briefly, whole EBs were washed once for 5 min in 1X PBS and fixed in 4% PFA in 1X PBS for 45 min at 4 °C. Following two washing steps of 5 min with 1X PBS, EBs were blocked in 10% donkey serum, 1% BSA, and 0.3% Triton-X100 in 1X PBS at room temperature for 1 hr. Whole EBs were incubated with the primary antibodies (Table 2) in 1% donkey serum (Jackson ImmunoResearch, West Grove, PA, USA), 1% BSA, and 0.1% Triton-X100 in 1X PBS for 4 days at 4 °C with gentle rocking. Following four washing steps of 5 min in 1X PBS, whole EBs were incubated with secondary antibodies (Table 3) overnight at 4 °C with continuous agitation. After four washing steps of 5 min in 1X PBS, EBs were incubated with 1 µg/mL DAPI in 1X PBS containing 0.3% Triton X-100 for 2 h at room temperature, and two washing steps in 1X PBS were performed prior to imaging.

### 4.4. Microscopy, Quantification, and Analysis

Brightfield images were collected using an EVOS microscope cell imaging system (Thermo Fisher Scientific), with 4×, 10×, and 20× objectives. For immunohistochemical analysis, samples were visualized using the Leica Inverted Confocal SP8 (Wetzlar, Germany). Each protocol was performed in triplicate, and three samples were collected at EB or RO stage. For cell counting analysis, five random regions of interest (ROIs) per section corresponding to one organoid were used. The ROI selection was restricted to neuroretina, compacted areas presenting laminar organization. Cells were counted using CellPose 2.0 using the model ‘cyto2′. Whole EBs were visualized using a Leica Thunder microscope system (D-35578 Wetzlar, Germany) with computational clearing.

### 4.5. PCR and Quantitative (q) PCR Analysis

RNA was isolated from pooled hESCs, EBs, neurospheres, and ROs (6–8 ROs randomly selected), using the Qiagen RNeasy Mini Kit (Qiagen, Valencia, CA, USA) according to the manufacturer’s instructions. cDNA was subsequently synthesized using oligo(dt) primer (Thermo Fisher Scientific) and Superscript III (Thermo Fisher Scientific) and PCR amplification using HOT FIREPol DNA Polymerase (Solis Biodyne, Tartu, Estonia). Primer sequences are listed in Table 4.

qPCR amplification was performed using the SensiFAST SYBR No-ROX Kit (Bioline, London, UK) on a CFX96 optical reaction module (BioRad CFX, CA, USA). The reaction parameters were as follows: 95 °C for 3 min, 95 °C for 15 s, 60 °C for 45 s (steps 2–3 are repeated 40 times), 95 °C for 30 s, 65 °C for 25 s, and 95 °C end. Plots show qPCR results as ΔΔCt. The results were normalized to *EF1α* (internal control) and then to the mean of the values on day 0 (on timeline representations). Values therefore show the amount of expression difference at each timepoint compared to day 0 (the earliest timepoint measured). Expression levels were plotted in GraphPad Prism 8.2.1 (Boston, MA, USA). Experiments were performed in triplicate and repeated three times.

### 4.6. Statistical Analysis

All data are represented as mean ± SD. Statistical analyses were performed using GraphPad Prism 8.2.1. The D’Agostino–Pearson test was used to test for normality. To compare the EB diameter, a Student’s *t*-test was performed. To compare the cell quantification between protocols at day 43, a Student’s *t*-test was performed. To compare the different timepoints between protocols for qPCR analysis, a Mann and Whitney test was performed (*** *p*  < 0.001, **** *p*  < 0.0001). The text and figure legends provide information about the number of samples for each experiment and condition.

## Figures and Tables

**Figure 1 ijms-25-01444-f001:**
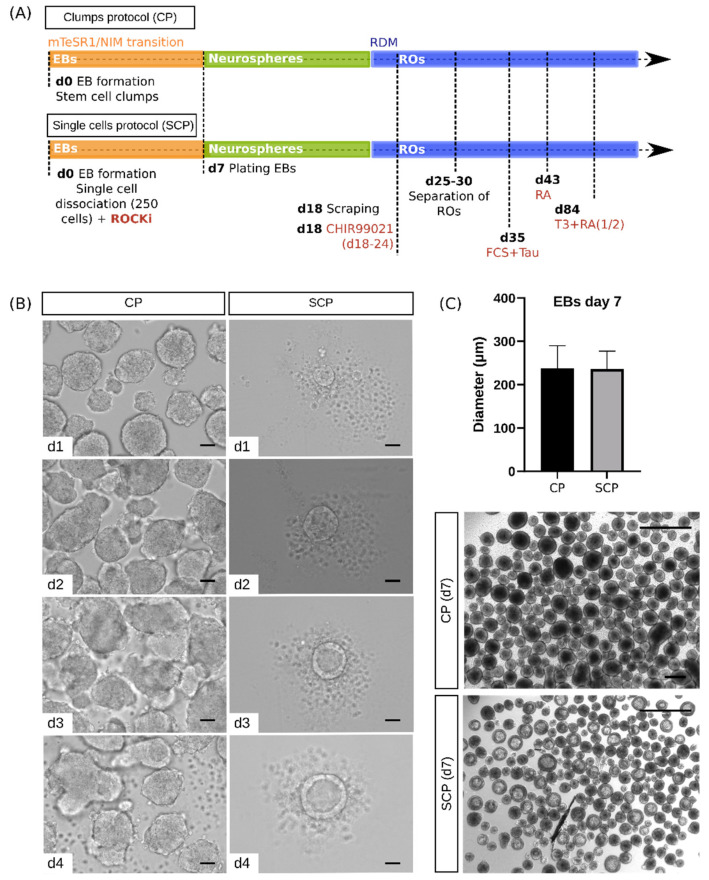
(**A**) Comparison of developmental timelines between different protocols. Both the clumps protocol (CP) (top) and the single cells protocol (SCP) (bottom) share many common steps. Both methods develop embryoid bodies (EBs) in a 3D environment over the course of 7 days, gradually transitioning from stem cell medium to neural induction medium (NIM) before being plated in a 2D environment to develop neurospheres. During this time, both protocols switch from NIM to retinal differentiation medium (RDM) at day 16, before being lifted and cultured in a 3D environment from day 18 onwards. Between days 18 and 24, 3 µM CHIR99021 is added. Retinal organoids (ROs) are then separated between days 25 and 30. For long-term suspension, the medium is supplemented with 10% Fetal Calf Serum (FCS), and 100 µM Taurine beginning on day 35. After a week, Retinoic Acid (1 µM) (RA) is added. At day 84, Triiodo-L-Thyronine (40 ng/ml) (T3) is added, and RA concentration is halved (supplements are highlighted in red). (**B**) Brightfield images of EBs during the first 4 days (d) of differentiation. Scale bars = 50 μm for both protocols on days 1, 2, 3, and 4. (**C**) Top: Average diameter of EBs ± SD on day 7 (*n* = 3 independent experiment, *n* = 50 EBs approximately per protocol); Student’s *t*-test. Bottom: Representative images of EBs used for diameter measurements. Scales bars = 1000 μm.

**Figure 2 ijms-25-01444-f002:**
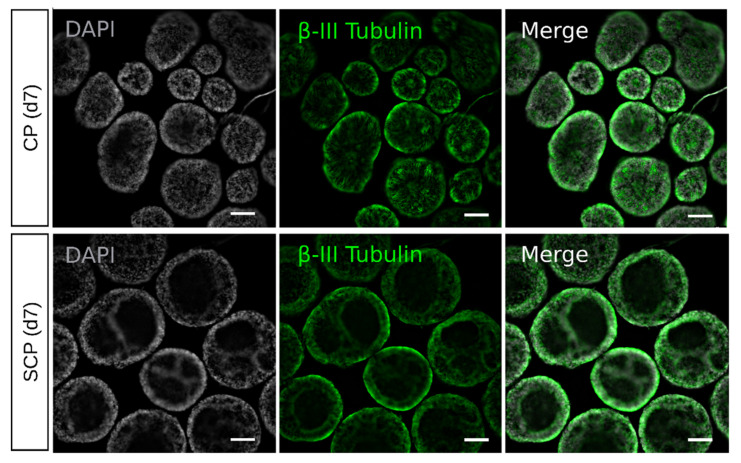
Whole-EB imaging. Confocal imaging of EBs generated using the clumps protocol (CP) (**top**) and the single cells protocol (SCP) (**bottom**) at day 7 showing DAPI-stained nuclei (in gray) and the pan-neuronal marker β-III Tubulin (in green). CP-derived embryoid bodies (EBs) display a dendritic distribution pattern of β-III Tubulin within the EB. SCP-derived EBs present β-III Tubulin expression in a tightly compacted later, highlighting the cavities within the EBs. Scale bars = 20 μm.

**Figure 3 ijms-25-01444-f003:**
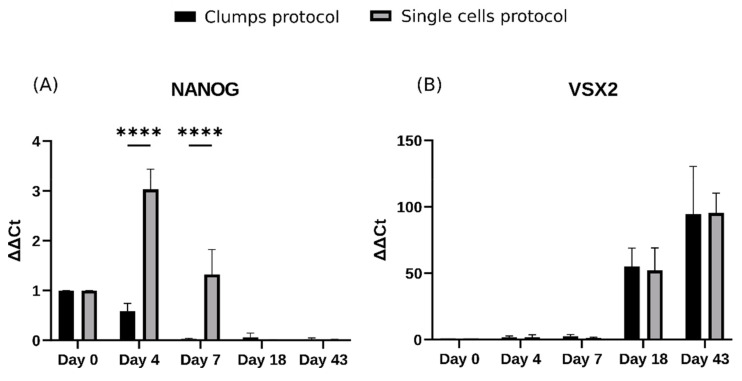
Temporal expression of the pluripotency gene *NANOG* and the neural progenitor *VSX2*. Material from the four primary representative stages during differentiation were collected at different timepoints: day 0 human embryonic stem cells (hESCs), days 4 and 7 embryoid bodies (EBs), day 18 (neurospheres), and day 43 retinal organoids (ROs). (**A**) Changes in gene expression for the pluripotency gene *NANOG*. (**B**) Changes in gene expression for the visual system homeobox 2 *VSX2*. Data are represented as mean ± SD; *n* = 3 independent experiments; **** *p*  < 0.0001; non-parametric Mann and Whitney test.

**Figure 4 ijms-25-01444-f004:**
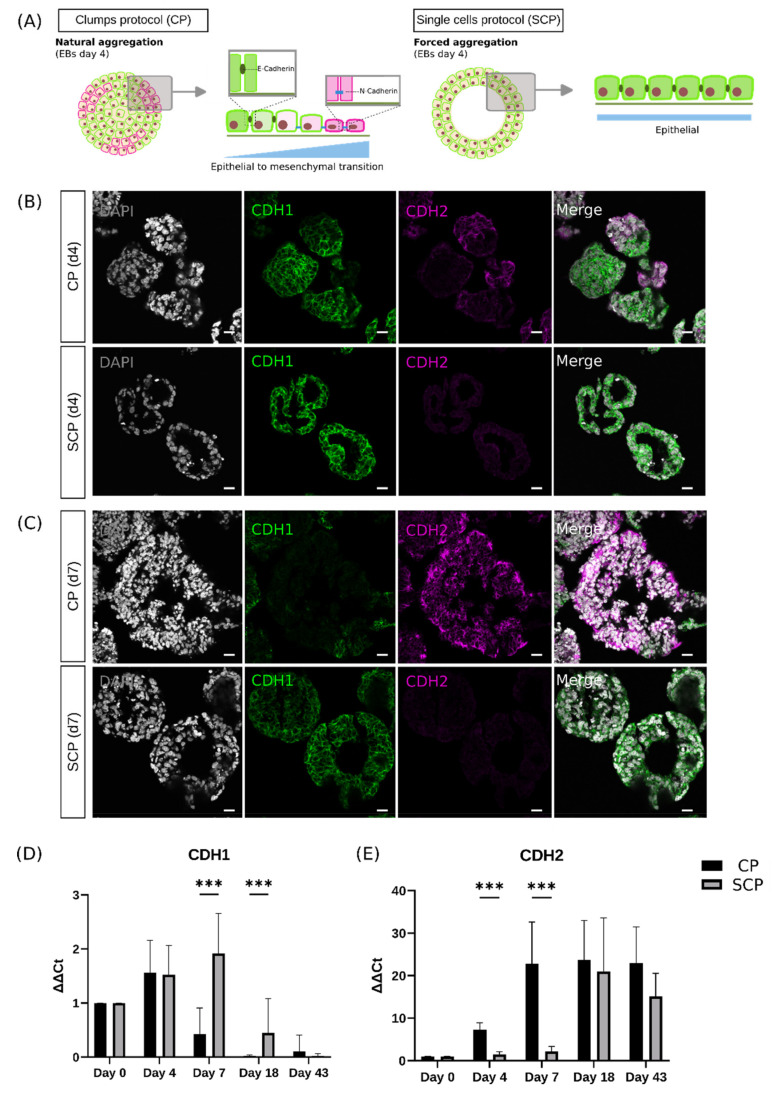
(**A**) Schematic representation of the E-cadherin to N-cadherin switch, a key event during the epithelial-to-mesenchymal transition. Epithelial cells undergo a loss of their characteristic polarity, break down cell–cell junctions, and gain migratory properties. Left: schematic representation of the E-cadherin to N-cadherin switch that clumps protocol (CP)-derived embryoid bodies (EBs) present. In these EBs, this transition can already be seen on day 4. Right: schematic representation of the E-cadherin to N-cadherin switch that single cells protocol (SCP)-derived EBs present. In these EBs, there is no indication of the transition by day 4 or 7. (**B**) Representative images of immunohistochemistry sections of EBs on day 4 (for both protocols) presenting DAPI-stained nuclei (in gray), E-cadherin marker CDH1 (in green), and N-cadherin marker CDH2 (in magenta). CP-derived EBs show scattered CDH1 distribution. In regions where CDH1 decreases, CDH2 expression is present. SCP-derived EBs present a high and uniform distribution of CDH1, with absence of CDH2. (**C**) Representative images of immunohistochemistry sections of EBs on day 7 (for both protocols) presenting DAPI-stained nuclei (in gray), CDH1 (in green), and CDH2 (in magenta). CP-derived EBs show a reduction of CDH1 expression, accompanied by a high CDH2 expression. SCP-derived EBs maintain a high and uniform distribution of CDH1, with absence of CDH2. Scale bars = 20 μm for both protocols on days 4 and 7. (**D**) Changes in gene expression for *CDH1.* (**E**) Changes in gene expression for *CDH2*. Significant differences for *CDH1* and *CDH2* are shown within the EBs’ stage on day 7. Data are represented as mean ± SD; *n* = 3 independent experiments; *** *p*  < 0.001; non-parametric Mann and Whitney test.

**Figure 5 ijms-25-01444-f005:**
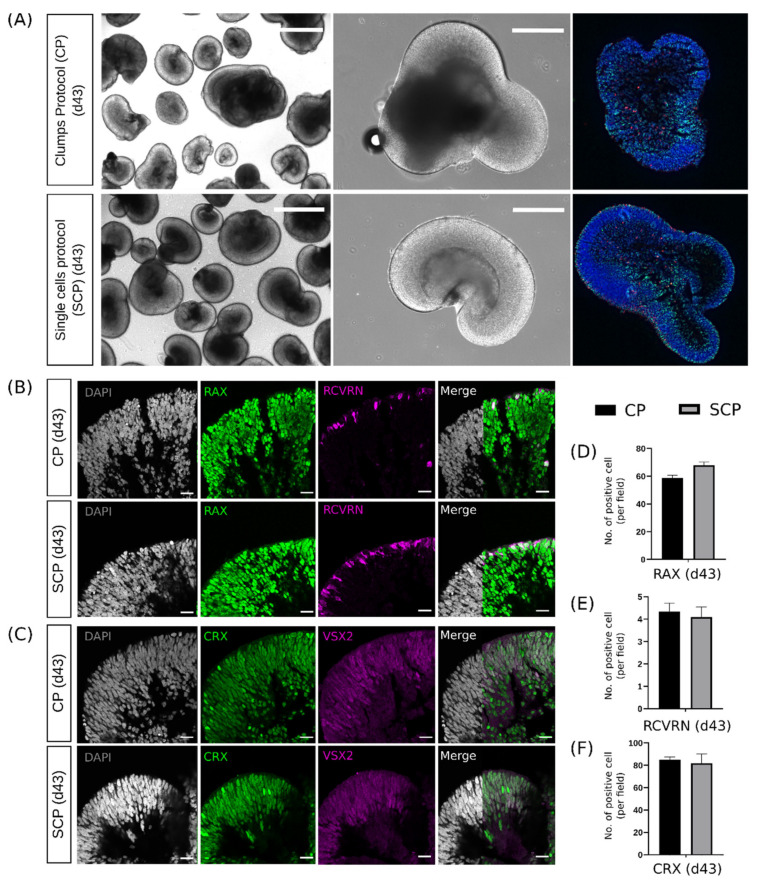
Representative images of retinal organoids (ROs) on day (d) 43. (**A**) Left: multiple ROs under a brightfield microscope after retinal and non-retinal separation step. Middle: detail of one RO with the representative neuroretina. Right: representative image of IHC section using the tile scan function. Scale bars = 1000 μm (left images) and (middle images). Both protocols (clumps protocol, CP and single cells protocol, SCP) generated ROs at day 43 with high heterogeneity in shape and size. (**B**) Representative images of the expression of DAPI-stained nuclei (in gray), the retinal homeobox protein RAX (in green), and the calcium-binding protein Recoverin RCVRN (in magenta) on day 43 for both protocols. These markers were robustly expressed within the neuroretinal regions, revealing similar architecture between protocols. (**C**) Representative images of the expression of DAPI-stained nuclei (in gray), the nuclear cone–rod homeobox protein CRX (in green), and the visual system homeobox 2 VSX2 (in magenta) on day 43 for both protocols. These markers were robustly expressed in both protocols, again revealing similar architecture between protocols. Scale bars = 20 μm. (**D**) Quantification of cell density of RAX in ROs on day 43 (*n* = 3 independent experiments, 3 ROs per experiment, 10–15 images per RO). (**E**) Quantification of cell density of RCVRN in 43-day ROs (*n* = 3 independent experiments, 3 ROs per experiment, 10–15 images per RO). (**F**) Quantification of cell density of CRX in 43-day ROs (*n* = 3 independent experiments, 3 ROs per experiment, 10–15 images per RO). Data are represented as mean ± SD; Student’s *t*-test. No significant differences were detected in the cell density quantification for RAX, RCVRN, and CRX.

**Figure 6 ijms-25-01444-f006:**
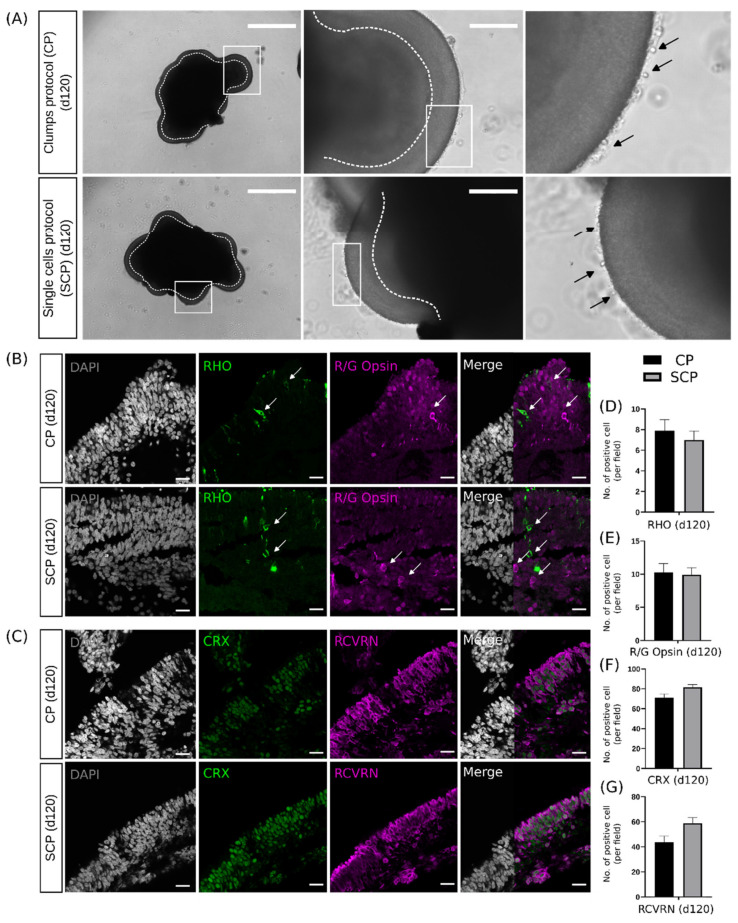
Representative images of retinal organoids (ROs) on day (d) 120. (**A**) Left: ROs under a bright field microscope (zoomed-in areas highlighted with white frames). Middle: detail of neuroretina (highlighted with dashed lines, zoomed-in areas highlighted with white frames). Right: protruding outer segments (highlighted with black arrows). ROs generated from both protocols (clumps protocol, CP and single cells protocol, SCP) increased their size. Scale bars = 1000 μm for pictures containing individual ROs (left images) and 200 μm for detail of individual ROs (middle images). (**B**) Representative images of the expression of DAPI-stained nuclei (in gray); the rod-specific rhodopsin, RHO, (in green); and the cone-specific red–green opsins, R/G opsin, (in magenta) on day 120 for both protocols (RHO and R/G opsin positive cells highlighted with white arrows). These photoreceptor-specific markers did not reveal differences in structural organization. (**C**) Representative images of the expression of DAPI-stained nuclei (in gray), the nuclear cone–rod homeobox protein CRX (in green), and the calcium-binding protein recoverin RCVRN (in magenta) on day 120 for both protocols. These markers reveal similar architecture between protocols. Scale bars = 20 μm. (**D**) Quantification of cell density of RHO in 120-day ROs (*n* = 1 experiment, 2 ROs, 10–15 images per RO), (**E**) Quantification of cell density of R/G opsin in 120-day ROs, (*n* = 1 experiment, 2 ROs, 10–15 images per RO) (**F**) Quantification of cell density of CRX in 120-day ROs, (*n* = 1 experiment, 2 ROs, 10–15 images per RO) (**G**) Quantification of cell density of RCVRN in 120-day ROs, (*n* = 1 experiment, 2 ROs, 10–15 images per RO). Data are represented as mean ± SD.

**Table 1 ijms-25-01444-t001:** Summary of previously published systematic studies. These studies were classified into the following categories: physical features of the microenvironment, exogenous signals, and starting cell conditions.

Author	Study Design			Cell Line	Category
Radojevic et al., 2021 [9]	Suspension protocol	Plating protocol		hiPSC	Physical features of the culture environment
Choy Buentello et al., 2022 [19]	U-bottom plate	V-bottom plate		hESC	Physical features of the culture environment
Berber et al., 2021 [18]	3D protocol	3D-2D-3D protocol	3D-2D-3D + BMP4 protocol	hiPSC	Physical features of the culture environment
Sanjurjo-Soriano et al., 2022 [10]	Without retinoic acid	With retinoic acid		hiPSC	Exogenous signals
Wagstaff et al., 2021 [12]	RI, IGF1, IWR1e, SB431542/LDN193189, CHIR99021, SU5402, CHIR99021/SU5402, DAPT			hESC	Exogenous signals
Chichagova et al., 2020 [20]	BMP4, IGF1			hiPSC	Exogenous signals
Mellough et al., 2019 [8]	Mechanical, enzymatic, or dissociation aggregation approaches to generate embryonic bodies			hESC, hiPSC	Starting cell conditions

**Table 2 ijms-25-01444-t002:** List of primary antibodies used for immunohistochemistry including name, dilution, catalog number, host, and manufacturer. Organized in alphabetical order.

Antibodies	Dilution	Catalog Number	Host	Manufacturer
CDH1	1:200	610182	Mouse	BD Biosciences (Becton, NJ, USA)
CDH2	1:200	22018-1-AP	Rabbit	Proteintech (Rosemont, IL, USA)
CRX	1:200	H00001406-M02	Mouse	Abnova (Taipei City, Taiwan)
GATA4	1:200	sc-25310	Mouse	Santa Cruz (Dallas, TX, USA)
RCVRN	1:200	AB5585	Rabbit	Merck Millipore (Burlington, MA, USA)
RHO	1:200	MAB5356	Mouse	Merck
R/G opsin	1:200	AB5405	Rabbit	Merck
SOX2	1:200	ab75627	Rabbit	Abcam (Cambridge, UK)
VSX2	1:200	ab16141	Sheep	Abcam

**Table 3 ijms-25-01444-t003:** List of secondary antibodies used for immunohistochemistry including name, dilutions, catalogue number, and manufacturer.

Antibodies	Dilution	Catalogue Number	Manufacturer
Donkey anti-mouse, Alexa Fluor 488	1:200	1509983	Jackson Immuno Research (West Grove, PA, USA)
Goat anti-mouse, Alexa Fluor 488	1:200	115546006	Jackson Immuno Research
Donkey anti-rabbit, Alexa Fluor 594	1:200	A21207	Thermo Fisher Scientific
Donkey anti-sheep, Alexa Fluor 594	1:200	A11016	Thermo Fisher Scientific

**Table 4 ijms-25-01444-t004:** Primer sequences and amplicon size in base pairs (bp). Organized in alphabetical order.

Gene	Forward	Reverse	Amplicon Size (bp)
*CDH1*	ATTTTTCCCTCGACACCCGAT	TCCCAGGCGTAGACCAAGA	109
*CDH2*	GTGCATGAAGGACAGCCTCT	TGGAAAGCTTCTCACGGCAT	138
*EF1α*	CAAAGCGACCCAAAGGTGGAT	AAATAAGCGCCGGCTATGCC	219
*NANOG*	TTTGGAAGCTGCTGGGGAAG	GATGGGAGGAGGGGAGAGGA	194
*VSX2*	CGGCGACACAGGACAATCTT	CCTCCAGCGACTTTTTGTGC	182

## Data Availability

Data are contained within the article and Appendix A.

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
