# Peer review of "The Analysis of Embryoid Body Formation and Its Role in Retinal Organoid Development"

_ijms, 2024, doi:10.3390/ijms25031444_

Round 1
Reviewer 1 Report
Comments and Suggestions for Authors
The purpose of this study was to investigate how the initial conditions of stem cells used in the formation of embryoid bodies (EBs) impact the differentiation and reproducibility of retinal organoids. The authors utilized a 3D floating culture method to create retinal organoids from stem cells. They compared two protocols: one using small clumps of stem cells from larger clones (CP) and another using an aggregation of single cells (SCP). The results of the study suggest that EBs generated with the SCP method maintain pluripotency for a longer time compared to those generated with the CP method. However, these differences decrease during development, potentially due to a compensatory mechanism during the 2D stage, resulting in similar neurospheres by day 18. Therefore, the initial differences observed in the EB stage do not negatively affect the gene expression or retinal architecture of the retinal organoids. The differences observed in the development of EBs generated by the two different protocols is not surprising because of the preexisting differences in the cell types used, hence the relevance of this study is not great. The manuscript is well written but the study design and the data presentation can be improved.
Specific concerns of the study are listed below.
1. The differences observed in the development of EBs generated by the two different protocols is not surprising because of the preexisting differences in the generation of SPC and SP, hence the relevance of this study is not great.
2. Only a limited no of assays were made to make a comparison between the two methods. The assays are limited to the expression of a few genes while several other critical genes showing differential expression may have been left unattended.
3. Although a compensatory mechanism theory is put forward for the differentiation process, gene expression evaluations were made only based on NANOG, VSX2, and CDH2 reaching comparable levels in neurospheres generated under both protocols.
4. Fig 1 C- this figure is not very relevant since the images show no difference between EBs generated by different protocols, consider for the supplementary figure.
5. Fig 5 D, this figure is not very relevant due to the absence of difference between groups, consider for the supplementary figure.
6. Fig 6- The ‘N’ used is very small to show meaningful statistical inference
7. The authors haven’t addressed and listed all the existing protocols for embryoid body formation. There are many other efficient and popular protocols. Is there a specific reason for the authors to choose the 2 different protocols listed in the manuscript?
8. Even though the authors have mentioned two different protocols they follow for embryoid body formation, it is not clear whether they followed the exact protocols post-embryoid body formation until the organoids reached the desired age. Did those papers follow the checkerboard method to dissociate RO in both of these protocols? This needs to be explained. If they deviate from the original protocol that can make a difference in results.
9. The authors have mentioned about “neurospheres”. The terminology ‘Neurospheres’ that appears throughout the manuscript is slightly confusing, is this the right terminology for these structures?
10. What is meant by ‘the neurospheres lifted’, explain more. Is this the same procedure as that of original protocol followed in this study? Any deviation need to be explained.
Comments on the Quality of English Languagenone
Reviewer 2 Report
Comments and Suggestions for Authors
The manuscript by Heredero Berzal et al is well-written and effectively illustrated. However, a few minor improvements can be made:
-
- Clearly define the meaning of "n" in each experiment for better understanding.
-
- Replace "rpm" with "rcf" to ensure consistent reporting of centrifugation conditions.
-
- Include organoid yield information for each protocol to provide insights into experimental efficiency.
Round 2
Reviewer 1 Report
Comments and Suggestions for Authors
manuscript has been modified in a satisfactorily